# Calcium Enrichment in Edible Mushrooms: A Review

**DOI:** 10.3390/jof9030338

**Published:** 2023-03-09

**Authors:** Zhen-Xing Tang, Lu-E. Shi, Zhong-Bao Jiang, Xue-Lian Bai, Rui-Feng Ying

**Affiliations:** 1School of Culinary Art, Tourism College of Zhejiang, Hangzhou 311231, China; 2School of Life and Environmental Sciences, Hangzhou Normal University, Hangzhou 311121, China; jiangbao_z@163.com (Z.-B.J.); baixl2012@163.com (X.-L.B.); 3College of Light Industry Science and Engineering, Nanjing Forestry University, Nanjing 210037, China; yingruifeng@hotmail.com

**Keywords:** calcium, enrichment, edible mushrooms, mechanism

## Abstract

Calcium is one of the essential minerals that enhances various biological activities, including the regulation of blood pressure, the prevention of osteoporosis and colorectal adenomas. Calcium-enriched edible mushrooms can be considered as one of the important daily sources of calcium in foods. Calcium accumulation in edible mushrooms is an effective way to enhance its activities because the organic state of calcium metabolites in edible mushrooms can be formed from the original inorganic calcium. The main calcium sources for calcium-enriched edible mushrooms’ cultivation are CaCO_3_, CaCl_2_ or Ca(NO_3_)_2_. The growth and metabolic process of edible mushrooms are significantly influenced by calcium enrichment. Generally, Ca at low levels is good for the production of edible mushrooms, whereas the reverse phenomenon for the growth of edible mushrooms at high Ca contents is observed. In addition, metabolites, for example, phenolics, flavonoids, polysaccharides, enzymes, minerals, etc., are improved when edible mushrooms are enriched at a moderate level of calcium. This review summarized the literature regarding the influence of calcium enrichment on edible mushrooms’ growth and major metabolites. Furthermore, the mechanisms of calcium enrichment in edible mushrooms were highlighted. Understanding calcium-enriched mechanisms in edible mushrooms would not only be beneficial to manipulate the cultivation of edible mushrooms having excellent biological activities and high levels of active Ca, but it would also contribute to the applications of calcium enrichment products in food industries.

## 1. Introduction

Calcium, one of the important microelements, has been demonstrated to exhibit great advantages to our health [1,2]. Calcium is not only essential for the growth of bones and teeth, but it also takes part in various physiological metabolisms of our body, such as the regulation of muscle contraction, blood coagulation, etc. [3,4,5]. Recently, the literature has offered findings that show the occurrence of various diseases, mainly osteoporosis, cardiovascular, male infertility, etc., is highly correlated to calcium deficiency [6,7,8]. So, the consumption of foods with high calcium levels is highly encouraged [9,10]. The intake of calcium from dairy sources, such as milk, cheese, yogurt, etc., is a commonly recommended way to satisfy the body’s calcium requirement [10,11]. Some non-dairy calcium sources from vegetables, including broccoli, kale, Chinese cabbage, etc., are also suggested [10]. The majority of people realize that calcium is a vital mineral for our health; however, humans are not getting sufficient calcium in their diets, in accordance with the recommendations of many nations and agencies [12,13,14,15]. To satisfy the need for calcium, the intake of calcium from calcium supplements is usually adopted. Many calcium supplements, including CaCl_2_, CaCO_3_, calcium gluconate, calcium amino acid chelate and peptide calcium, have been made available, which have played a great role in providing calcium to the human body [16,17]. However, these calcium supplements have some drawbacks, for instance, the low solubility of inorganic calcium, poor absorption and utilization efficiency, etc. [17,18]. Therefore, to meet the calcium requirements, the development of higher-bioavailability, safer and cheaper calcium supplements is rather necessary. 

Many beneficial microorganisms such as probiotics, yeasts, edible mushrooms, etc., show great potential for the accumulation of minerals [17,19,20,21]. Among these microorganisms, edible mushrooms can be regarded as an interesting object of supplementation. In numerous countries, edible mushrooms have been part of the daily diet for several thousand years [22,23]. They are regarded as a nutritious food, referring to the many nutritious substances they contain, including polysaccharides, minerals, dietary fibers, proteins, vitamins, etc. [23,24]. Furthermore, owing to their many bioactive characteristics, for example, anti-cancer, anti-bacterial, anti-oxidation, anti-inflammatory, etc., edible mushrooms can also be considered as functional foods with potential advantages for our health [24,25,26,27,28,29]. There are about 100 species that are commercially available in the global mushroom market. Around 20 species have the potential to be cultivated at an industrial level [30,31]. The world outputs of edible mushrooms have increased dynamically year by year, varying from 10.5 million tons in 2016 to 11.8 million tons in 2019, with an increase of around 57% over the last 10 years [32]. It is important to highlight that the worldwide market for edible mushrooms in 2019 was USD 16.9 billion in 2019, whereas it is anticipated to reach USD 19.04 billion by 2026 [33]. China is the largest manufacturer of edible mushrooms in the world, and its production is still on the rise [34,35,36,37]. 

Owing to edible mushrooms’ excellent capacity to accumulate minerals, numerous studies on minerals enriched in edible mushrooms have been carried out to help improve the nutritive value of edible mushrooms. Edible mushrooms fortified with calcium are extremely interesting, showing great potential as a calcium dietary supplement [38,39,40]. In view of the increasing demand for natural dietary supplements, Ca-fortified edible mushrooms can be regarded as a type of marketable product with great commercial potential. Compared to studies of other enriched minerals such as selenium [41,42,43], the investigations for Ca accumulation in edible mushrooms are relatively limited. The most common calcium enrichment method involves the addition of exogenous calcium salts into a substrate or fermentation medium. Consequently, calcium-fortified edible mushrooms have the potential to be a safe and effective source of daily Ca supplementation, exhibiting the benefits of safety and effectively promoting organic Ca formation [44,45,46]. To develop edible mushroom foods containing high Ca levels and good biological activities, this paper has reviewed the recent pertinent literature on Ca enrichment in edible mushrooms, with a focus on well-investigated edible mushroom species and their Ca metabolites. Furthermore, the mechanisms for calcium enrichment in edible mushrooms were also highlighted. It is the first review paper focused on calcium enrichment in edible mushrooms. This review will build the basis for future investigations on Ca accumulation within edible mushrooms. 

## 2. Effects of Various Factors on Calcium Enrichment in Edible Mushrooms

Edible mushrooms are rich in many essential minerals [47,48,49,50,51], including potassium, calcium, phosphorus, and magnesium, which are often deficient in our daily diet [52,53]. Accordingly, the investigation of Ca-enriched edible mushrooms has been a growing research area. Through the incorporation of Ca into active biomacromolecules during the metabolic process, the mycelium and the fruiting bodies of edible mushrooms are able to convert inorganic-state Ca to organic-state Ca, which has higher bioavailability and is safer compared to the inorganic form [38,54]. Several studies have been conducted to investigate the capacity of edible mushrooms, including *Pleurotus eryngii*, *Lentinula edodes*, *Hypsizygus marmoreus*, *Pholiota nameko* and *Ganoderma lucidum*, to accumulate calcium from a variety of Ca sources [55,56,57,58,59,60]. Ca content (Table 1) in edible mushrooms highly depends on several factors, for example, edible mushroom species, growing environments, etc. [50]. Edible mushrooms, for instance, *Flammulina velutipes* [60], *P. ostreatus*, *H. marmoreus*, *Auricularia auricula* [61,62], *Coprinus comatus*, are excellent calcium-enriched candidates (Table 1). Generally, the total calcium is lower in edible mushrooms than in vegetables [63,64]. In an effort to enrich edible mushrooms with calcium, Tabata and Ogura found that the Ca level in fruiting bodies of *H. marmoreus* was improved as potato sucrose agar (PSA) and sawdust media were added with 1.0% Ca salts [65]. Choi et al. determined the calcium-enriching effect of *P. eryngii* in sawdust medium with a supplement of calcined starfish powder [66]. These authors also expected that numerous environmental factors, such as pHs, moisture concentrations, climate conditions, etc., could have an additional influence on calcium accumulation within the fruiting bodies of edible mushrooms [66]. In addition, the abilities of *P. ostreatus* and *P. nameko* to accumulate calcium in PSA and sawdust media have also been well characterized [59,67].

As one of the typical edible mushrooms, *P. eryngii* is acknowledged as an antioxidant resource, containing a large number of beneficial compounds and secondary metabolites, which may prevent oxidative damage [68]. It is also regarded as a high-efficiency calcium accumulator and can change inorganic calcium into organic calcium [69,70]. Akyuz et al. found that *P. eryngii* tended to have higher mineral accumulations, because the Mg and Ca contents in fruiting bodies were higher than other minerals [71]. Similarly, increased Ca content (14.94 mg/100 g) was observed in *P. eryngii* cultured on rice straw [72]. These differences in the calcium contents of *P. eryngii* were also attributed to the different culture media used or different substrate components. Moreover, the wide variation in the Ca content of *P. eryngii* grown on different media was similar to previous investigations [73,74,75]. In 2023, He et al. investigated the influence of five kinds of exogenous calcium sources (calcium chloride, calcium amino acid chelate, calcium lactate, calcium nitrate and calcium carbonate) on *P. eryngii* mycelia and fruiting bodies and found the optimum exogenous calcium (calcium lactate) could improve the yield of *P. eryngii* fruiting bodies and shorten its growth cycle [69]. However, in the investigation of Bu et al., the authors found different edible mushroom species (*Pleurotus nebrodensis*, *P. eryngii* and *Pleurotus citrinopileatus*) showed a significant effect on calcium enrichment. In addition, *P. nebrodensis* was a more suitable Ca-enriched edible mushroom candidate compared to other kinds of edible mushrooms [18].

In general, the main Ca metabolic products present in edible mushrooms are in an organic state. The distribution of Ca metabolites in edible mushrooms differs according to the cultivated cultivar and growing conditions. Specifically, 62.4% of Ca was combined with protein in *Cordyceps sinensis*, and the polysaccharide fraction contained 11.5% of Ca. A total of 80.5% of inorganic Ca was transferred into organic Ca [20,59]. The calcium enrichment of *Laetiporus sulphureus* showed similar findings. The degree of organic calcium reached 85.85% when the calcium content was 100 mg/L [54]. However, for *Poria cocos*, although 97.91% calcium was absorbed, only 24.57% organic calcium was detected [76,77]. 

Although edible mushrooms are excellent at accumulating Ca and can be grown over a wide range of Ca levels, their abilities to accumulate Ca differ from cultivar to cultivar and with culturing conditions, Ca sources and dosages (Table 2). Particularly, Ca sources and doses can highly affect Ca enrichment in edible mushrooms (Table 2). Current studies on Ca accumulation in edible mushrooms principally use CaCO_3_, CaCl_2_ and Ca(NO_3_)_2_ as Ca sources [78]. *F. velutipes* is one type of popular food in China due to its excellent anti-cancer and immunostimulating abilities [60,79,80]. Fan et al. showed that with the addition of 1~2% light CaCO_3_ and 1~2% shellac, the mycelia of *F. velutipes* grew denser, and the output and the quality of fruiting bodies improved [60,80]. In addition, in support of these results, it has been shown that adding 0.5% CaCO_3_ into potato sucrose agar (PSA) medium slightly enhanced the mycelium growth of *H. marmoreus*, while adding 5.0% CaCO_3_ into the same medium resulted in total inhibition [65]. However, it was observed that adding Ca phosphate and Ca carbonate into sawdust media did not affect the growth of *P. eryngii* cultivated on both potato dextrose agar (PDA) and sawdust media with a supplement of Ca salts, while adding CaSO_4_ inhibited the growth of mycelium [81].

Different sources of calcium are commonly used in the commercial production of *Agaricus* spp. Thus, calcium sulfate (gypsum) is used as an ingredient in mushroom compost formulations and is applied in the early stages of the composting process, mainly for colloid flocculation, making the compost less greasy, improving aeration and subsequently facilitating mycelial growth [85,86,87,88,89]. Spent lime obtained in the production of sugar from sugar beet, consisting mainly of calcium carbonate, is used as ingredient of casings. The technical interest in the use of spent lime is basically due to its buffering capacity and its ability to improve the casing layer structure, giving casing soil a more or less dense texture [90,91,92]. Other sources of calcium have been evaluated in casings for the production of *Agaricus subrufescens* [93]. Calcium chloride can be used in irrigation water to improve the quality of fruit bodies, mainly their texture and dry matter content [94,95,96,97,98,99]. Irrigation with calcium lactate solutions has also been proposed [94]. The dipping of mushrooms in solutions of calcium chloride, calcium lactate and calcium nitrate has been evaluated in order to preserve the quality and increase the postharvest life of button mushrooms [100].

Inedible Ca sources have been used, such as agricultural lime, starfish powder, eggshells, oyster shells etc., which contain CaCO_3_ as the major component [101,102]. Accordingly, the bioconversion of inedible calcium sources is a good method for utilizing these renewables [64]. Zhang et al. found the mycelia of *H. marmoreus* grew more densely when 3.0% light CaCO_3_ or 3.0% shell powder was added into the medium [103]. In addition, for calcium enrichment in *P. eryngii*, Choi et al. found that supplementing sawdust medium with 1.0% oyster shell powder did not inhibit the mycelium growth of *P. eryngii*. The addition of 2.0% oyster shell powder into sawdust medium potentially elevated the calcium level within the fruiting bodies of *P. eryngii* up to 315.7 ± 15.7 mg/100 g, without prolonging the duration of spawning run, and delaying the days to primordial production. However, adding over 4.0% oyster shell powder into the sawdust medium resulted in the significant suppression of mycelial growth [101]. Furthermore, in Choi’s group, the authors found that the Ca level within the fruiting bodies of *P. eryngii* was improved through calcined starfish powder treatment. Supplementing the sawdust medium with 1.0% starfish powder did not inhibit the mycelial growth of *P. eryngii* and elevated the calcium level up to 256.0 ± 16.3 mg/100 g within the fruiting bodies of *P. eryngii* without prolonging the spawning period and delaying the occurrence of primordial germination [66]. These findings demonstrated that the development of calcium-fortified edible mushroom foods could be achieved using inedible calcium sources. 

Typically, low Ca content stimulates the growth of edible mushrooms, while a high level of Ca suppresses the growth of mycelia and can even cause toxicity, with the feature of declined biomass and decomposed cells in edible mushrooms. The reason was that low Ca contents might activate enzymes in edible mushrooms, whereas high Ca contents might inhibit enzyme activity in the mycelia [104]. *H. marmoreus*, known as the Jade mushroom, exhibits many advantages to our health, including immunity-boosting, cancer-fighting and aging-preventing properties [57,103]. The mycelium growth of *H. marmoreus* was promoted at low Ca contents (500~2000 mg/L) but inhibited at higher contents (>2000 mg/L) [105]. Likewise, Sun et al. demonstrated that adding 60 mg/L CaCl_2_ into PDA medium promoted the mycelium growth of *H. marmoreus*. The effect of 50~100 mg/L calcium content on the mycelial growth rate was not significant, while at high concentrations, CaCl_2_ significantly inhibited the mycelial growth [106]. Interestingly, the optimal growth and calcium enrichment of *G. lucidum* was achieved when Ca(NO_3_)_2_ (600 mg/100 g) was added into the medium. The calcium enrichment of *G. lucidum* was significantly reduced when the addition level exceeded 800 mg/100 g [60,107]. Furthermore, Ca contents (0~2.0 g/L) did not inhibit the mycelium growth of *Wolfiporia cocos*. Calcium enrichment in the mycelia was as high as 89.11 mg/g [76]. A similar growth phenomenon has also been observed in *P. ostreatus* [108], *L. edodes* [102] and *C. comatus* [109]. 

Organic and inorganic Ca salts affect edible mushroom growth in different ways. In general, edible mushrooms are more responsive to organic Ca salts. Qin et al. investigated the influence of four kinds of calcium sources (CaCO_3_, CaCl_2_, Ca(NO_3_)_2_ and amino acid calcium) on the calcium accumulation ability of *G. lucidum* and found that the strongest ability to accumulate calcium was observed with 0.2% Ca(NO_3_)_2_ or when amino acid calcium was added. The amount of enriched calcium in *G. lucidum* reached 584.13 mg/100 g [60,110]. Similar results were observed for *L. edodes.* Chen et al. found that all calcium compounds (CaCO_3_, calcium lactate, CaSO_4_, CaCl_2_ and Ca(NO_3_)_2_) except calcium nitrate had a significant promoting effect on mycelial growth, and calcium sulfate was most advantageous to mycelial growth, whereas calcium lactate, as a result, was the most suitable as a calcium source to enrich calcium in mycelia [111]. Furthermore, the combination of calcium salts was also adopted to enrich calcium in *C. sinensis*. With the combination of calcium sources (40% Ca(NO_3_)_2_ + 60% CaCO_3_) at a total Ca^2+^ addition of 3.0 g/L, the biomass of *C. sinensis* could reach as high as 32.1 g/L [20]. 

Cultivation methods also have an impact on Ca accumulation and the growth of edible mushrooms. In particular, edible mushrooms are capable of tolerating higher Ca levels when grown in solid substrate versus liquid culture, which may be caused by the slow rate of mass exchange and extended incubation period. Under solid culture conditions, no significant effect of any of calcium levels studied on the mycelium growth of *H. marmoreus* could be observed, while under liquid culture conditions, the mycelial growth of *H. marmoreus* was inhibited when the calcium content was higher than 250 mg/L. The authors indicated that great differences in the calcium tolerance of *H. marmoreus* to solid and liquid culture conditions might be attributed to the cultivation state of the mycelium [57]. Similarly, Xiong et al. examined the influences of different calcium contents on the mycelial growth and calcium content of *G. lucidum* after solid and liquid cultivation. Compared to solid cultivation, a higher calcium content in *G. lucidum* (100.6 mg/100 g) was obtained when the calcium content in the liquid culture was 200 μg/mL [60,82]. Furthermore, Chen et al. reported that when the Ca^2+^ content was 1000~9000 mg/L in the solid culture, the mycelial growth was promoted. When the Ca^2+^ content was over 12,000 mg/L, mycelium growth was inhibited. However, in the liquid culture, the mycelian polysaccharide level was the highest at a Ca^2+^ content of 5000 mg/L [79]. In 2023, He et al. observed that the growth of *L. sulphureus* was better compared to that of *Poriacocos*, *Armillaria* and *Monascus* in solid cultivations. When the addition of calcium dose was 100 mg/L in the liquid culture, the biomass and calcium content of *L. sulphureus* were significantly higher than those of *Poriacocos*, *Armillaria* and *Monascus* [54,77]. Conversely, in the study by Yang et al., the mycelial growth of *L. edodes* 4754 was significantly promoted in the presence of ≤8608.5 mg/L CaCl_2_ during solid cultivation, whereas the mycelial growth was stimulated with higher calcium content (17,217.0 mg/L CaCl_2_) during liquid cultivation [56]. The reasons for these great differences need to be investigated further.

## 3. Effect of Calcium Enrichment on Nutritive Value of Edible Mushrooms

Growing edible mushrooms and using them for applications in foods and pharmaceuticals are rapidly growing research fields [112,113,114,115,116]. The mycelium is one of the most valuable sources for the bioactives believed to have health advantages [113,115,117,118,119]. Mineral-enriched edible mushrooms drive us to optimize culture media to produce the most healthful mycelia. This, in turn, could make it possible to produce food products or dietary supplements with functional properties [113,120,121]. The addition of Ca ions into a medium might not only increase their enrichment in mycelia (Table 3) but also affect metabolite production [113,118]. 

### 3.1. Polysaccharides

Polysaccharide synthesis is a complicated metabolic pathway that involves a large number of enzymes [124,125]. Edible mushrooms may respond differently to changes in the environment depending on the stages of growth. Generally, during the early enrichment stage, no significant dynamic variations in mycelium growth and the accumulation of polysaccharides are recorded. With the increase in cultivation time, the maximum mycelial biomass and content of polysaccharides were found [55,82,124]. In the work carried out by Ji et al., the influences of four kinds of mineral ions (calcium, magnesium, zinc and copper) on *P. djamor* polysaccharide were investigated. The authors found that the order of effects of four divalent mineral ions on the production of *P. djamor* polysaccharide was the following: magnesium ion > copper ion > zinc ion > calcium ion [122]. Adil et al. studied the impact of calcium enrichment on the polysaccharide production of *L. edodes*. During the enrichment process, Ca^2+^ induction increased the polysaccharide level [124]. These findings demonstrated that the minerals showed varying impacts on the growth and metabolism of various edible mushrooms. Ca^2+^ was able to suppress the mycelium growth of *Tricholoma mongolicum* while promoting the production of polysaccharides [59,126]. In addition, the mycelial growth and yield were promoted when 1000 mg/L Ca(NO_3_)_2_ was added in the medium. The highest concentration of 41.72% for crude fiber was obtained in *Inonotus-obliquus*-enriched calcium, which increased by 2.59% over the control group [123]. In 2020, Chen et al. studied the influence of the addition of Ca ions on polysaccharide content in the mycelia of *F. velutipes*. With Ca^2+^ content of 5000 mg/L, the mycelial polysaccharide levels were highest, exceeding the control group by 139.2% [60,79]. 

### 3.2. Phenolics and Flavonoids 

Phenolics and flavonoids are responsible for numerous features which are related to free radical scavenger activity [127,128]. The addition of phenolic and flavonoid compounds into foods is highly recommended because they can enhance the nutritional value of foods [51]. An increase in the mineral contents of the medium is associated with the greater enrichment of phenolic and flavonoid compounds present in edible mushrooms [129]. Adding minerals into the medium improved the contents of phenolic and flavonoid compounds within the fruiting bodies of *H. erinaceus*, *G. lucidum*, *A. aegerita* and *C. indica*, resulting in superior antioxidant characteristics [60,130,131]. Selenium accumulation showed a significant effect on the antioxidant metabolite pattern of the fruiting bodies of *C. indica*, and adding 5.0 μg/mL Sn improved the level of total phenolics, whereas the level of total phenolics decreased at higher doses of Sn addition [131]. Similarly, some of the mixtures of Fe and Ca at various contents significantly affected phenolic acids’ synthesis. Supplementing Fe and Ca mixture modified the profile, stimulating the synthesis of some new constituents, thus significantly increasing the content of phenolic. In addition, the contents of total phenolic and flavonoids were also highest in Ca-fortified *P. nameko.* However, remarkable variations in the phenolic component were not observed in *P. nameko* fortified with Fe and Ca [59,129]. The authors thought that the variation in phenolic and flavonoid levels in fortified edible mushrooms probably resulted from activating or deactivating the synthesis pathway for phenolic and flavonoids at various stages, which might have been attributed to the types of calcium salt in the enriched medium [129]. 

### 3.3. Enzymes

Edible mushrooms are able to secrete numerous exo-enzymes, including laccase, cellulase, xylanase, amylase and others, during the growth process. These extracellular enzymes can decompose biomacromolecules such as cellulose, protein, nucleic acid, etc., into small molecules, which provide nutrients to the mycelium and the fruiting bodies of edible mushrooms [104,132]. It was shown that 4.0 mg/mL Ca^2+^ in the medium was suitable content to enrich calcium in *L. edodes*. With the increase in calcium ion content, the esterase isoenzyme activity of *L. edodes* decreased significantly [133]. However, Ca^2+^ addition enhanced the phospho-glucose isomerase and phosphoglucomutase enzyme activity in *L. edodes*, whereas Na^+^ increased UDP-glcpyrophosphorylase activity [124].

### 3.4. Other Bioactive Compounds

After being treated by exogenous calcium, *P. eryngii* showed a strong accumulation ability for calcium. After treatment with calcium lactate, the total soluble sugars and soluble proteins within the fruiting bodies of *P. eryngii* were firstly improved, followed by a decrease as the calcium lactate contents increased [69]. However, the impacts of different calcium lactate contents on the levels of fat and free amino acids within the fruiting bodies of *P. eryngii* were not apparent. Similarly, supplementing all Ca sources reduced the levels of K and P in *P. ostreatus* but generally increased the concentrations of Mg, Na, Si, Cl and S [64]. Additionally, adding 10 mM Mn^2+^ and Ca^2+^ enhanced the production of total ganoderic acid by 2.2- and 3.7 times, respectively [134,135]. These findings suggest that the nutritional value of edible mushrooms could be significantly improved through calcium enrichment.

## 4. The Mechanisms of Calcium Enrichment in Edible Mushrooms

Although Ca accumulation in edible mushrooms has been well documented, only a few studies have documented the underlying mechanisms for Ca enrichment, which are highly related to Ca absorption, transport and metabolic process. Therefore, investigating Ca metabolic processes in edible mushrooms will help us to understand Ca enrichment mechanisms at the molecular level. Once key genes and enzymes involved in Ca metabolic processes are elucidated, we can manipulate Ca accumulation in edible mushrooms, which will benefit the development of Ca-accumulating edible mushrooms. 

The pathway of mineral enrichment is different in various edible mushrooms. The absorption of minerals is mainly through the mycelia of edible mushrooms. Afterwards, the minerals are transported across the plasma membrane in passive ways, driven by different electrochemical potentials. Additionally, the minerals can also be transported into the cell of edible mushrooms in active ways, depending on the carriers on the protoplasmic membrane, through expending energy to cross the protoplasmic membrane. Ozean et al. concluded that edible mushrooms are enriched with minerals mainly through active transport ways and that edible mushrooms are able to accumulate higher contents of minerals compared to green plants [136]. Regarding calcium enrichment in edible mushrooms, with the increase in calcium ion content, Ca^2+^-ATPase, a calcium transport system on the cell membrane of the mycelia, was activated accordingly. After that, the calcium absorption and transport rate was improved progressively. However, when the addition of calcium ion was increased to the threshold value, Ca^2+^-ATPase activity was inhibited, which caused a decline in calcium absorption and transport, resulting in poor calcium accumulation in the mycelia [69]. Therefore, the appropriate increase in exogenous calcium stimulated the growth of edible mushrooms’ mycelia. Lee et al. thought that the passive transport of calcium through a nonspecific channel in the plasma membrane and further diffusion through the mycelium or extracellular transportation from the medium via the intermycel cavity into the fruiting bodies could be possible. However, Lee et al. pointed out that the importance of calcined oyster shell powder involved in either of these transport methods obviously required further examination [81]. Similarly, Bu et al. also found that the normal accumulation of calcium was achieved with the help of Ca^2+^-ATPase [18] due to the fact that calcium ions could not freely pass through the hydrophobic membrane of edible mushrooms. In 2022, Zhang et al. showed that edible mushrooms mainly absorb calcium ions through mycelia, and calcium ions are mainly transported into the inside of cells in active transport ways with the help of plasma membrane carriers [57]. Calcium ions entering edible mushrooms would bind to reactive groups of biomacromolecules such as protein, polysaccharide, nucleic acid, etc., and form organismal inclusions or chelates, thus completing the bio-transformation of inorganic calcium into the organic state [137]. In plants, the uptake of Ca is dependent on Ca^2+^ form and level, as well as on high-affinity or low-affinity membrane transporter activity [138,139]. However, whether Ca carriers in edible mushrooms are the same as in plants, and whether there are general or specific carriers in edible mushrooms, is still not clear.

Extracellular polymeric substances (EPSs) produced by edible mushrooms also can adsorb and transform calcium ions. They are polysaccharides attached to the surface of edible mushrooms’ mycelia or surrounding the mycelia and are essential for maintaining cellular morphology, secreting extracellular enzymes and resisting external disturbances [45].

## 5. Conclusions

Ca-enriched edible mushrooms can be regarded as one of the important daily Ca supplements. A number of investigations have been performed regarding Ca enrichment in edible mushrooms. Ca enrichment not only has an effect on the growth of edible mushrooms but is also involved in the metabolism of bioactives. In this review, we mainly focused on the Ca enrichment ability of edible mushrooms, the enrichment process and the effect of enrichment on metabolites. However, detailed information regarding Ca absorption and metabolic methods is relatively limited. Hence, more fundamental investigations need to be carried out before we can regulate Ca enrichment in edible mushrooms without adversely affecting their functions. Many genes and enzymes participate in Ca enrichment. Through the extensive study of key genes and enzymes during the Ca enrichment process, we can not only manipulate the metabolic process of Ca-enriched compounds, but also synthesize Ca-enriched compounds with good bioactivities or functions. The focus of future studies should be mapping metabolic pathways for calcium accumulation and the discovery of the key genes and enzymes involved in calcium enrichment. Through these studies, it is expected that a vast number of value-added edible mushrooms with high Ca contents will represent a feasible choice as dietary calcium supplements.

## Figures and Tables

**Table 1 jof-09-00338-t001:** Calcium level in some edible mushrooms.

Edible Mushrooms	Levels (μg/g)	References
*Agrocybe aegerita*	203.99 ± 6.47	Lin et al., 2015 [48]
*Flammulina velutipes*	890.34 ± 17.80	
*Hypsizygus marmoreus*	1279.12 ± 25.58	
*Lentinus edodes*	840.39 ± 5.23	
*Pleurotus eryngii*	796.03 ± 15.92	
*Agaricus blazei m* *urrill*	425.81-703.79	Liu et al., 2018 [49]
*Agrocybe cylindracea*	115.21-564.40	
*Auricularia auricula*	1971.83-6103.99	
*Coprinus comatus*	1014.67-1874.72	
*Cyptotrama chrysopeplum*	58.25-259.83	
*Dictyophora indusiata*	86.35-475.48	
*Flammulina velutipes*	32.62-164.09	
*Hericium erinaceus*	13.49-25.23	
*Lentinus edodes*	154.96-650.09	
*Pholiota nameko*	282.76-740.47	
*Pleurotus eryngii*	23.29-47.50	
*Pleurotus ostreatus*	681.56-1143.17	
*Tremella fuciformis*	88.03-691.07	
*Volvariella volvacea*	925.59-1613.94	

**Table 2 jof-09-00338-t002:** Calcium enrichment in some edible mushrooms.

Edible Mushrooms	Calcium Source	Optimized Ca Content for Enrichment (mg/L *)	Calcium-Enriched Amount (mg/100 g **)	References
*Ganoderma lucidum*	CaCl_2_	200	100.6	Lee et al., 2006 [82]
*Hypsizygus marmoreus*	CaCl_2_	100	2239.8	Zhang et al., 2022 [57]
*Inonotus obliquus*	Ca(NO_3_)_2_	1000	21	Yu et al., 2016 [83]
*Pleurotus* *nebrodensis*	CaCl_2_	6000	790.6	Bu et al., 2009 [18]
*Pleurotus* *ostreatus*	CaCl_2_	6000	491.67	He et al., 1998 [84]
*Poria cocos*	CaCl_2_	2000	89.11	Wang et al., 2007 [76]

* mg/L of cultivation medium, ** mg/100 g of dry mycelium weight.

**Table 3 jof-09-00338-t003:** Effect of calcium enrichment on nutritive value of edible mushrooms.

Edible Mushrooms	Results	References
*P* *leurotus eryngii*	Calcium enrichment improved the total soluble sugars and protein in fruiting bodies, whereas calcium accumulation did not show a significant impact on fat and free amino acids in fruiting bodies.	He et al., 2020 [69]
*P* *leurotus djamor*	The maximum crude polysaccharide content was obtained as calcium content was varied from 0.05 to 0.10 mg/mL	Ji et al., 2017 [122]
*G* *anoderma lucidum*	Calcium enrichment could improve the content of extracellular polysaccharide.	Xiong et al., 2009 [82]
*Inonotus obliquus*	The highest crude fiber content of 41.72% in calcium-enriched sample was observed. Furthermore, the highest total triterpene content of 0.058 mg/mL was obtained under calcium-enriched conditions.	Yu et al., 2016 [83] Guo et al., 2015 [123]
*Flammulina velutipes*	The highest polysaccharide content was achieved with Ca^2+^ content of 5000 mg/L, whereas polysaccharide accumulation was inhibited with Ca^2+^ content of 12,000 mg/L. Additionally, the amylase activity was the highest with Ca^2+^ content of 1000 mg/L.	Chen et al., 2020 [79]
*Laetiporus sulphureus*	Dentate acid content of 18.34 mg/g was obtained when calcium content in liquid culture was 100 mg/L.	He et al., 2023 [54]

## Data Availability

No applicable.

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
