# Peer review of "Calcium Enrichment in Edible Mushrooms: A Review"

_jof, 2023, doi:10.3390/jof9030338_

Round 1

Reviewer 1 Report

The manuscript presents a review of different aspects related to calcium enrichment and its effect in the case of edible mushrooms. It is an original work that is well documented and updated with an interesting proposal and high significance and impact. It presents, in my opinion, sufficient depth of analysis, generic value and scientific level to be published, with minor corrections, in Journal of Fungi.

The inclusion in section 2 of additional text related to different sources of calcium enrichment applied in the production of Agaricus spp. is considered of interest. As an orientation, a text that can be used including the accompanying references is included below. An additional literature review can be carried out. It may also be of interest to mention the use of calcium sulfate and calcium carbonate in spawn production.

Proposed additional text:

Different sources of calcium are commonly used in the commercial production of Agaricus spp. Thus, calcium sulfate (gypsum) is used as an ingredient in mushroom compost formulations and applied in the early stages of the composting process, mainly for colloid flocculation, making the compost less greasy, improving aeration and subsequently facilitating mycelial growth (Gerrits, 1988; Colak, 2004; Pardo et al., 2007; Kariaga et al., 2012; Pardo-Giménez et al., 2020). Spent lime obtained in the production of sugar from sugar beet, consisting mainly of calcium carbonate, is used as ingredient of casings. The technical interest in the use of spent lime is basically due to its buffering capacity and its ability to improve the casing layer structure, giving the casing soil a more or less dense texture (Visscher, 1988; Pardo-Giménez et al., 2017; Dias et al., 2021). Other sources of calcium have been evaluated in casings for the production of Agaricus subrufescens (Zied et al., 2012). Calcium chloride can be used in irrigation water to improve the quality of fruitbodies, mainly their texture and dry matter content (Miklus and Beelman, 1996; Hartman et al., 2000; Diamantopoulou and Philippoussis 2001; Philippoussis et al., 2001; Chikthimmah et al., 2005; KaÅ‚użewicz et al., 2014). Irrigation with calcium lactate solutions has also been proposed (KaÅ‚użewicz et al., 2014). Dipping of mushrooms in solutions of calcium chloride, calcium lactate and calcium nitrate has been evaluated in order to preserve the quality and increase the postharvest life of button mushroom (Khademi and Khoveyteri-Zadeh, 2022)

New references:

Gerrits, J.P.G. (1988). Nutrition and compost. In The Cultivation of Mushrooms; van Griensven, L.J.L.D., Ed., Interlingua T.T.I. Ltd.: East Grinstead, Sussex, UK, 1988; pp. 29-72.

Colak, M. Temperature profiles of Agaricus bisporus in composting stages and effects of  different composts formulas and casing materials on yield. Afr. J. Biotechnol. 2004, 3(9), 456-462. https://doi.org/10.5897/AJB2004.000-2089

Pardo, A.; Perona, M.A.; Pardo, J. (2007). Indoor composting of vine by-products to produce substrates for mushroom cultivation. Span. J. Agric. Res. 2007, 5(3), 417-424. https://doi.org/10.5424/sjar/2007053-260

Kariaga, M.G.; Nyongesa, H.W.; Keya, N.C.O.; Tsingalia, H.M. Compost physico-chemical factors that impact on yield in button mushrooms, Agaricus bisporus (Lge) and Agaricus bitorquis (Quel) Saccardo. Journal of Agricultural Sciences 2012, 3(1), 49-54. https://doi.org/10.1080/09766898.2012.11884685

Pardo-Giménez, A.; Pardo, J.E.; Dias, E.S.; Rinker, D.L.; Caitano, C.E.C.; Zied, D.C. Optimization of cultivation techniques improves the agronomic behavior of Agaricus subrufescens. Scientific Reports 2020, 10, 8154. https://doi.org/10.1038/s41598-020-65081-2

Visscher, H.R. (1988). Casing soil. In The Cultivation of Mushrooms; van Griensven, L.J.L.D., Ed., Interlingua T.T.I. Ltd.: East Grinstead, Sussex, UK, 1988; pp. 73-89.

Pardo-Giménez, A.; Pardo, J.E.; Zied D.C. Casing materials and techniques in Agaricus bisporus cultivation In Edible and Medicinal Mushrooms: Technology and Applications. Zied, D.C., Pardo-Giménez, A., Eds., Wiley-Blackwell: Chichester, UK, 2017; pp. 149-174. https://doi.org/10.1002/9781119149446.ch7

Dias, E.S.; Zied, D.C.; Pardo-Giménez, A. Revisiting the casing layer: casing materials and management in Agaricus mushroom cultivation. Cienc. Agrotec. 2021, 45, e0001R21. https://doi.org/10.1590/1413-70542021450001R21

Zied, D.C.; Pardo-Giménez, A.; Minhoni, M.A.; Villas, R.L.; Álvarez-Ortí, M.; Pardo-González, J.E. Characterization, feasibility and optimization of Agaricus subrufescens growth based on chemical elements on casing layer. Saudi J. Biol. Sci. 2012, 19, 343-347. https://doi.org/10.1016/j.sjbs.2012.04.002

Miklus, M.B.; Beelman, R.B. CaCl2 treated irrigation water applied to mushroom crops (Agaricus bisporus) increases Ca concentration and improves postharvest quality and shelf life. Mycologia 1996, 88(3), 403-409. https://doi.org/10.1080/00275514.1996.12026667

Hartman, S.C.; Beelman, R.B.; Simons, S. Calcium and selenium enrichment during cultivation improves the quality and shelf life of Agaricus mushrooms. Mushroom Sci. 2000, 15(2), 499-505.

Diamantopoulou, P.; Philippoussis, A. Production attributes of Agaricus bisporus white and off-white strains and the effect of calcium chloride irrigation on productivity and quality. Sci. Hortic. 2001, 91(3-4), 379-391. https://doi.org/10.1016/S0304-4238(01)00274-6

Philippoussis,A.; Diamantopoulou, P.; Zervakis, G. Calcium chloride irrigation influence on yield, calcium content, quality and shelf-life of the white mushroom Agaricus bisporus. J. Sci. Food Agric. 2001, 81, 1447-1454. https://doi.org/10.1002/jsfa.968

Chikthimmah, N.; La Borde, L.F.; Beelman, R.B. Hydrogen peroxide and calcium chloride added to irrigation water as a strategy to reduce bacterial populations and improve quality of fresh mushrooms. J. Food Sci. 2005, 70(6), 273-278. https://doi.org/10.1111/j.1365-2621.2005.tb11446.x

KaÅ‚użewicz, A.; Górski, R.; Sobieralski, K.; Siwulski, M.; Sas-Golak, I. The effect of calcium chloride and calcium lactate on the yielding of Agaricus bisporus (Lange) Imbach. Ecol. Chem. Eng. S 2014, 21(4), 677-683. https://doi.org/10.1515/eces-2014-0049

Khademi, O.; Khoveyteri-Zadeh, S.  Determination the best source of calcium for button mushroom conservation. J. Hortic. Postharvest Res. 2022, 5(2), 177-186. https://doi.org/10.22077/jhpr.2022.4797.1246

Other minor fixes:

L68: Se (selenium) instead of Sn

Include the full name (only genus and species) in tables 1, 2 and 3 to avoid confusion and facilitate independent reading of the tables. For example, P. is used for both Pleurotus and Pholiota in table 1.

Modify: Agrocybe aegerita instead of Agrocube aegirit in table 1.

L241: Remove NN22.

L242-244: Check that the name P. centrale is correct and include the bibliographical reference in the sentence where it appears. I have not been able to identify the species in dictionaries of fungi or in speciesfungorum (https://www.speciesfungorum.org/Names/Names.asp).

References section: Use bold font in year of reference 28; complete reference 51 (year of publication, 2022?); complete reference 102 (volume, pages)

Author Response

Dear Reviewer

Thank you for your feedback. According to your suggestion, we corrected the manuscript carefully. Now, if you have any questions about the corrected one, please contact me.

Wait for your further information

Best regards

Tang, Zhen-Xing

Reviewer 1#

The manuscript presents a review of different aspects related to calcium enrichment and its effect in the case of edible mushrooms. It is an original work that is well documented and updated with an interesting proposal and high significance and impact. It presents, in my opinion, sufficient depth of analysis, generic value and scientific level to be published, with minor corrections, in Journal of Fungi.

The inclusion in section 2 of additional text related to different sources of calcium enrichment applied in the production of Agaricus spp. is considered of interest. As an orientation, a text that can be used including the accompanying references is included below. An additional literature review can be carried out. It may also be of interest to mention the use of calcium sulfate and calcium carbonate in spawn production.

Answer: Good suggestion! The additional text as you suggested, was added into the paper. Please see the paper.

Proposed additional text:

Different sources of calcium are commonly used in the commercial production of Agaricus spp. Thus, calcium sulfate (gypsum) is used as an ingredient in mushroom compost formulations and applied in the early stages of the composting process, mainly for colloid flocculation, making the compost less greasy, improving aeration and subsequently facilitating mycelial growth (Gerrits, 1988; Colak, 2004; Pardo et al., 2007; Kariaga et al., 2012; Pardo-Giménez et al., 2020). Spent lime obtained in the production of sugar from sugar beet, consisting mainly of calcium carbonate, is used as ingredient of casings. The technical interest in the use of spent lime is basically due to its buffering capacity and its ability to improve the casing layer structure, giving the casing soil a more or less dense texture (Visscher, 1988; Pardo-Giménez et al., 2017; Dias et al., 2021). Other sources of calcium have been evaluated in casings for the production of Agaricus subrufescens (Zied et al., 2012). Calcium chloride can be used in irrigation water to improve the quality of fruit bodies, mainly their texture and dry matter content (Miklus and Beelman, 1996; Hartman et al., 2000; Diamantopoulou and Philippoussis 2001; Philippoussis et al., 2001; Chikthimmah et al., 2005; KaÅ‚użewicz et al., 2014). Irrigation with calcium lactate solutions has also been proposed (KaÅ‚użewicz et al., 2014). Dipping of mushrooms in solutions of calcium chloride, calcium lactate and calcium nitrate has been evaluated in order to preserve the quality and increase the postharvest life of button mushroom (Khademi and Khoveyteri-Zadeh, 2022).

New references:

Gerrits, J.P.G. Nutrition and compost. In The Cultivation of Mushrooms; van Griensven, L.J.L.D., Ed., Interlingua T.T.I. Ltd.: East Grinstead, Sussex, UK, 1988; pp. 29-72.

Colak, M. Temperature profiles of Agaricus bisporus in composting stages and effects of different composts formulas and casing materials on yield, Afr. J. Biotechnol. 2004, 3, 456-462.

Pardo, A.; Perona, M.A.; Pardo, J. Indoor composting of vine by-products to produce substrates for mushroom cultivation. Span. J. Agric. Res2007, 5, 417-424.

Kariaga, M.G.; Nyongesa, H.W.; Keya, N.C.O.; Tsingalia, H.M. Compost physico-chemical factors that impact on yield in button mushrooms, Agaricus bisporus (Lge) and Agaricus bitorquis (Quel) Saccardo. J. Agric. Sci2012, 3, 49-54.

Pardo-Giménez, A.; Pardo, J.E.; Dias, E.S.; Rinker, D.L.; Caitano, C.E.C.; Zied, D.C. Optimization of cultivation techniques improves the agronomic behavior of Agaricus subrufescensSci. Rep2020, 10, 8154.

Visscher, H.R. Casing soil. In The Cultivation of Mushrooms; van Griensven, L.J.L.D., Ed., Interlingua T.T.I. Ltd.: East Grinstead, Sussex, UK, 1988; pp. 73-89.

Pardo-Giménez, A.; Pardo, J.E.; Zied D.C. Casing materials and techniques in Agaricus bisporus cultivation In Edible and Medicinal Mushrooms: Technology and Applications; Zied, D.C., Pardo-Giménez, A., Eds., Wiley-Blackwell: Chichester, UK, 2017; pp. 149-174.

Dias, E.S.; Zied, D.C.; Pardo-Giménez, A. Revisiting the casing layer: casing materials and management in Agaricus mushroom cultivation. Cienc. Agrotec2021, 45, e0001R21. 

Zied, D.C.; Pardo-Giménez, A.; Minhoni, M.A.; Villas, R.L.; Álvarez-Ortí, M.; Pardo-González, J.E. Characterization, feasibility and optimization of Agaricus subrufescens growth based on chemical elements on casing layer. Saudi J. Biol. Sci2012, 19, 343-347.

Miklus, M.B.; Beelman, R.B. CaCl2 treated irrigation water applied to mushroom crops (Agaricus bisporus) increases Ca concentration and improves postharvest quality and shelf life. Mycologia 1996, 88, 403-409.

Hartman, S.C.; Beelman, R.B.; Simons, S. Calcium and selenium enrichment during cultivation improves the quality and shelf life of Agaricus mushrooms. Mushroom Sci2000, 15, 499-505.

Diamantopoulou, P.; Philippoussis, A. Production attributes of Agaricus bisporus white and off-white strains and the effect of calcium chloride irrigation on productivity and quality. Sci. Hortic2001, 91, 379-391.

Philippoussis,A.; Diamantopoulou, P.; Zervakis, G. Calcium chloride irrigation influence on yield, calcium content, quality and shelf-life of the white mushroom Agaricus bisporusJ. Sci. Food Agric2001, 81, 1447-1454.

Chikthimmah, N.; La Borde, L.F.; Beelman, R.B. Hydrogen peroxide and calcium chloride added to irrigation water as a strategy to reduce bacterial populations and improve quality of fresh mushrooms. J. Food Sci2005, 70, 273-278.

KaÅ‚użewicz, A.; Górski, R.; Sobieralski, K.; Siwulski, M.; Sas-Golak, I. The effect of calcium chloride and calcium lactate on the yielding of Agaricus bisporus (Lange) Imbach. Ecol. Chem. Eng. S2014, 21, 677-683.

Khademi, O.; Khoveyteri-Zadeh, S.  Determination the best source of calcium for button mushroom conservation. J. Hortic. Postharvest Res2022, 5, 177-186. 

Other minor fixes:

L68: Se (selenium) instead of Sn

Answer: The revision was made.

Include the full name (only genus and species) in tables 1, 2 and 3 to avoid confusion and facilitate independent reading of the tables. For example, P. is used for both Pleurotus and Pholiota in table 1.

Modify: Agrocybe aegerita instead of Agrocube aegirit in table 1.

Answer: The revision was made. Please see the paper.

L241: Remove NN22.

Answer: The revision was made. Please see the paper.

L242-244: Check that the name P. centrale is correct and include the bibliographical reference in the sentence where it appears. I have not been able to identify the species in dictionaries of fungi or in species fungorum (https://www.speciesfungorum.org/Names/Names.asp).

Answer: Wrong name for this fungi, and delete it from the manuscript.

References section: Use bold font in year of reference 28; complete reference 51 (year of publication, 2022?); complete reference 102 (volume, pages)

Answer: The revision was made. Please see the paper.

Reviewer 2 Report

Calcium Enrichment in Edible Mushrooms: A Mini Review:  Zhen-Xing Tang , Lu-E Shi and Rui-Feng Ying

Note: Agaricus bisporus  represents world-wide about 11% of total production (Status and trends in world mushroom production-III -World Production of Different Mushroom Species in 21st Century (fao.org).  So has it been deliberated left out or are the authors focusing on non-Agaricus species?  The title needs changing to reflect the focus of the review.

Note: There are several references with “for peer review” in this journal.  Will they be properly referenced in the published version?

Note: The authors use time sensitive words like “recently” (5 times).  This is only valid for a short time.  Hopefully this publication will be used much later than “recent”.  If the authors want to call attention to the year, then they can say “In 2022 Ji et al.  ….”

Note: The references have various species names misspelled.  Some I have noted.  Perhaps, there are others.

LL 18-19 – I presume it is production.  However, this needs to be clarified.

L 21  “can …. were”  (verb tense agreement) - change to “were improved … were”

L 26 -  “conduced ….”  - unclear -rewrite to the end of the sentence.

L 31 – “certified that it can”  change to “demonstrated to”

L 32 -  “keep” change to “assist”

L 35 – delete “many” ; change ‘have’ to ‘has’

L 36 – delete ‘including’; change ‘relative’ to “correlated’

L 38 -  yogurt not with ‘h’

L 40 – delete ‘the’

L 42 – after ‘our health’ add ‘;’ not ‘,’

L 50 -  move ‘such as … etc’ after ‘microorganisms’

L 52 – rewrite ‘In …history’  like ‘In numerous countries, edible mushrooms have been part of the daily diet for several thousand years’

L 54 – change ‘own’ to ‘owing’; change ‘bioactives’ to ‘bioactive’

L 55 – delete ‘contributing…excellent’

L 57 -  unclear – are there 100 species commercially available  or is that a potential number?

LL 57 – 58 -  unclear – are the 20 species at an industrial level now or has the technology been developed sufficiently to move to an industrial scale?

L 60 – delete ‘cultivation marketing’ ; change to ‘market’

L 62 – change ‘the production’ to ‘its production’; delete ‘of the rest… worldwide’

L 66 – insert ‘with’ between ‘fortified’ and ‘calcium’

L 68 – reference for “selenium” only includes non-Agaricus.  Is this intentional?  Robert Beelman at the Pennsylvania State University was the leader in enriching Agaricus bisporus and others with selenium.

L 73 – delete ‘was conducted to review”; change ‘ the’ to ‘this’;  add ‘has reviewed’

L77 – delete’ which …. industry’

L 80 -  change ‘are enriched’  to ‘are rich’

L 81 – do references ’46 -50’ reference diet deficiencies or the actual minerals.  If minerals, move to appropriate location.

L 82 -  delete ‘hot’ and ‘gradually’

L 84 – change ‘safety’ to ‘safe’  I think that a brief explanation or phrase as to why the organic Ca is more save than inorganic is in order.

Note: Ref 52 – Ganoderma not Gamoderma; check elsewhere in the References

L 85-85  -  I don’t think the authors need to add the shorting of the Latin names in parenthesis.  Normal usage will shorten the genera to the first letter of its name after the first usage.  As well line 89 is not consistent with the practice in these lines.

L 87 – delete ‘as showed in Table 1’  this is wordy;  add after “content” as “(Table 1) as done in line 90.

Ref 57 spelling of ‘ostreatus’; same in 60 and 61 (advise check whole manuscript)

L 92 – ‘was added’ to ‘were added’

L 94 – 96 -  “these findings …” – this statement is not justified by the previous references.

Ref 58: ‘peck’ should be “Peck”

L 98 –  The word “distribution” to me signifies the various parts of the mushroom, eg., pileus, stipe, lamella, etc. so  … I suggest “Calcium level in edible mushrooms”

L 98 – the table.  (1) There is no logic to the position of the names or levels.  I would suggest alphabetize edible mushrooms.  (2) some names are with abbreviations and others not.  I would like to see all names in full (3) a few names have the describer following it.  Inconsistent.  (4) references:  as it is the reader does not know if 47 continues to 48 or what???

L 100 and others -  starting a sentence with the abbreviated scientific name is not recommended by many style references.  Check with the editor. 

L 101 – “used” this infers a purpose.  What is it?

L 102 -  “et al” is italic here and elsewhere no.  What is the correct style?

L 104 – ‘compared to .. contents’  -- unclear

L 110  - ‘short’ to ‘shorten’

L 111 – why describer “Sing”  and not describers used in others.  Inconsistent.

L 114 -  “at” to ‘in an’

L 115 – ‘differs’ to ‘differs’  - agreement with distribution in singular

L 117 – ‘sulfur fungus’   - use scientific name

L 119 -  ‘obtained’  ? able to be extracted from the tissue??

L 129 -  ‘affect on the growth’  --  delete the ‘on’

L 132 -  Table 2 .  (1) mushrooms are not in any order (2) Please spell out the genera (3) Optimized Ca content need some explanation in footnote; contents are in 4 units (4) Calcium enriched needs explanation in footnote – mg/100 g of what?? (5) I would change title ‘ … in some edible mushrooms’

L 135 – 136  ‘provides’ to ‘provide’ --- agreement with singular subject

L 143 – ‘same group’ -  unclear

L 145 -  “might’  -  words shows doubt.  Is this speculation?

L 146 –‘demonstrate’ to ‘demonstrated’

L 156 ‘could’ – is this speculation or fact? ; same with L 159, L 171

L 160 – ‘pattern’  -  unsure of the references (not bibliography) in the text

L 162 -3 – ‘in general … mushrooms’ should it not read ‘Edible mushrooms are more responsive to organic Ca salts’ ???

L 165 ‘exhibited the strongest ability’  - rephrase

L 185, 188 (2022) or (2017)  – authors have switched referencing style; Sulfur fungus – Latin name please

L 188 – ‘study of Yang’ – change ‘of’ to ‘by’

L 194 – ‘are one of rapidly …’ to ‘are rapidly growing …’

L 196 -  change ‘impelled us’ ‘impells us’  or ‘drives us’ or ‘stimulates us’

L 198 – “As showed in Table 3,”  wordy; place word (Table 3) in the sentence

L 199 – 200 -  I don’t think the authors need this sentence.

L 201 -   Why this table?  Why not incorporate it into a separate paragraph that begins with “The addition of Ca ions …”

L 205 – “in different” to “differently”

L 207 – delete “not”  - “not” is not required if “no” is earlier

L 207 – ‘the maximum of mycelium’ to ‘maximum mycelial”

L 208 – ‘are’ to ‘were’

L 208 – ‘In the work studied by Ji et al. (2017),’  - inconsistent reference style

L 215 – delete ‘obviously’

L 221 – “scavenge free radicals activity”  -  I think the authors want to say “free radical scavenger activity”

L 222, 223 “Normally”, “Not surprising”  - wordy, delete

L 224 – “normally”  -  this usage prepares the reader for a contrast like “however”.  I don’t see it.  So delete the word.

L 226 – starting sentence with element abbreviation is not recommended.  Use full word – Selenium

L 230 – “could” – is this conjecture? Probably authors are saying “mixture modified”

L 233 – ‘phenolics’ to ‘phenolic’

L 335 – unclear relationship or reference to something earlier

L 240   ‘was in favourable’   - unclear

L 251 ‘could reduce” to ‘reduced’

L 252 ‘initially’  - this word presupposes a contrast.  I see nothing.  Delete it.

L 253 – ‘suggested’ to ‘suggest’

L 257 – ‘literature .. studied’ to ‘have documented’

L 258 – ‘with’  to ‘to’

L 259 – ‘therefore, the illustration of’ to ‘Investigating the’

L 259 – ‘about’ to ‘of’

Section 4 “The mechanisms ….  needs to be re-organized (re-written).  Paragraph 1 mostly is research desired.  There are English flow issues and grammar in it.  It cannot be cut and pasted directly elsewhere.  Paragraph 2 is a review as stated for the paper.

L 264 – ‘mushrooms, afterwards, ‘ to “mushrooms. Afterwards,’  -  run-on sentence

L 267 – reference style issue

L 269 – 271 – rewrite into two sentences and clarify

L 272 – ‘was improved’ to ‘was increased’

L 273 -4 – unclear

L 275 -  reference style problem

L 277 -  “the authors”  to “these authors” for clarity

L 279 – reference style problem

L 279 -  what is the antecedent of “it” – needs clarification

L 280 –  this is a run-on sentence.  Divide into two.

L 281 – recently problem.  Also reference style problem.

L 281 – ‘absorbed’ to ‘absorb’

L 282 -  ‘under’ to ‘with’

L 283 – don’t start sentence with abbreviation, no comma after mushrooms.

L 290 – don’t start sentence with abbreviation

L 298  - ‘regarding on’ to ‘regarding’

L 300 – 302 – unclear; rewrite

L 305  ‘choose’ to ‘choice’

Author Response

Dear Reviewer

Thank you for your feedback. According to your suggestion, we corrected the manuscript carefully. Now, if you have any questions about the corrected one, please contact me.

Wait for your further information

Best regards

Tang, Zhen-Xing

Reviewer 2#

Note: Agaricus bisporus represents world-wide about 11% of total production (Status and trends in world mushroom production-III-World Production of Different Mushroom Species in 21st Century (fao.org).  So has it been deliberated left out or are the authors focusing on non-Agaricus species?  The title needs changing to reflect the focus of the review.

Answer: Good suggestion! The additional information on different calcium sources applied in the production of Agaricus spp, was added into the paper. Please see the paper.

Note: There are several references with “for peer review” in this journal. Will they be properly referenced in the published version?

Answer: The reference section was corrected again. Please see the paper.

Note: The authors use time sensitive words like “recently” (5 times).  This is only valid for a short time. Hopefully this publication will be used much later than “recent”.  If the authors want to call attention to the year, then they can say “In 2022 Ji et al.  ….”

Answer: The revision was made. Please see the paper.

Note: The references have various species names misspelled. Some I have noted. Perhaps, there are others.

LL 18-19 – I presume it is production. However, this needs to be clarified.

Answer: The revision was made.

L 21  “can …. were”  (verb tense agreement) - change to “were improved … were”

Answer: The revision was made.

L 26 -  “conduced ….”  - unclear -rewrite to the end of the sentence.

Answer: The revision was done.

L 31 – “certified that it can”  change to “demonstrated to”

Answer: The revision was made.

L 32 -  “keep” change to “assist”

Answer: The revision was made.

L 35 – delete “many” ; change ‘have’ to ‘has’

Answer: The revision was made.

L 36 – delete ‘including’; change ‘relative’ to “correlated’

Answer: The revision was made.

L 38 -  yogurt not with ‘h’

Answer: The revision was made.

L 40 – delete ‘the’

Answer: The revision was made.

L 42 – after ‘our health’ add ‘;’ not ‘,’

Answer: The revision was made.

L 50 -  move ‘such as … etc’ after ‘microorganisms’

Answer: The revision was made.

L 52 – rewrite ‘In …history’  like ‘In numerous countries, edible mushrooms have been part of the daily diet for several thousand years’

Answer: The revision was made according to your suggestion.

L 54 – change ‘own’ to ‘owing’; change ‘bioactives’ to ‘bioactive’

Answer: The revision was made.

L 55 – delete ‘contributing…excellent’

Answer: The delete was made.

L 57 -  unclear – are there 100 species commercially available or is that a potential number?

Answer: The clarification was made.

LL 57 – 58 -  unclear – are the 20 species at an industrial level now or has the technology been developed sufficiently to move to an industrial scale?

Answer: The clarification was made.

L 60 – delete ‘cultivation marketing’ ; change to ‘market’

Answer: The revision was made.

L 62 – change ‘the production’ to ‘its production’; delete ‘of the rest… worldwide’

Answer: The revision was made.

L 66 – insert ‘with’ between ‘fortified’ and ‘calcium’

Answer: The revision was made. 

L 68 – reference for “selenium” only includes non-Agaricus.  Is this intentional?  Robert Beelman at the Pennsylvania State University was the leader in enriching Agaricus bisporus and others with selenium.

Answer: The reference was added.

  1. David, A.W.; Robert, B.B.; David, M.B. Manganese and other micronutrient additions to improve yield of Agaricus bisporus. Bioresource Technol. 2006, 97, 1012-1017.

L 73 – delete ‘was conducted to review”; change ‘ the’ to ‘this’;  add ‘has reviewed’

Answer: The revision was made.

L77 – delete’ which …. industry’

Answer: The revision was made.

L 80 -  change ‘are enriched’  to ‘are rich’

Answer: The revision was made.

L 81 – do references ’46 -50’ reference diet deficiencies or the actual minerals. If minerals, move to appropriate location.

Answer: The revision was made. New references were added.

Cormick, G.; Belizan, J.M. Calcium intake and health. Nutrients 2019, 11, 1606.

Vavrusova, M.; Skibsted, L.H. Calcium nutrition. Bioavailability and fortification. LWT-Food Sci. Technol. 2014, 59, 1198-1204.

L 82 -  delete ‘hot’ and ‘gradually’

Answer: The revision was made.

L 84 – change ‘safety’ to ‘safe’  I think that a brief explanation or phrase as to why the organic Ca is more save than inorganic is in order.

Answer: The revision was made according to your suggestion.

Note: Ref 52 – Ganoderma not Gamoderma; check elsewhere in the References

Answer: The revision was made.

L 85-85  -  I don’t think the authors need to add the shorting of the Latin names in parenthesis. Normal usage will shorten the genera to the first letter of its name after the first usage. As well line 89 is not consistent with the practice in these lines.

Answer: The revision was made according to your suggestion.

L 87 – delete ‘as showed in Table 1’  this is wordy;  add after “content” as “(Table 1) as done in line 90.

Ref 57 spelling of ‘ostreatus’; same in 60 and 61 (advise check whole manuscript)

Answer: The revision was made.

L 92 – ‘was added’ to ‘were added’

Answer: The revision was made.

L 94 – 96 -  “these findings …” – this statement is not justified by the previous references.

Answer: The expression for this sentence was changed.

Ref 58: ‘peck’ should be “Peck”

Answer: The revision was made.

L 98 –  The word “distribution” to me signifies the various parts of the mushroom, eg., pileus, stipe, lamella, etc. so  … I suggest “Calcium level in edible mushrooms”

Answer: The revision was made according to your suggestion.

L 98 – the table.  (1) There is no logic to the position of the names or levels.  I would suggest alphabetize edible mushrooms.  (2) some names are with abbreviations and others not.  I would like to see all names in full (3) a few names have the describer following it. Inconsistent.  (4) references: as it is the reader does not know if 47 continues to 48 or what???

Answer: Good suggestion! The table 1 was carefully corrected according to your suggestion.

L 100 and others - starting a sentence with the abbreviated scientific name is not recommended by many style references.  Check with the editor. 

Answer: The revision was made.

L 101 – “used” this infers a purpose.  What is it?

Answer: The revision was made.

L 102 -  “et al” is italic here and elsewhere no.  What is the correct style?

Answer: The revision was made.

L 104 – ‘compared to .. contents’  -- unclear

Answer: It was removed.

L 110  - ‘short’ to ‘shorten’

Answer: The revision was made.

L 111 – why describer “Sing”  and not describers used in others.  Inconsistent.

Answer: It was removed.

L 114 -  “at” to ‘in an’

Answer: The revision was made.

L 115 – ‘differs’ to ‘differs’  - agreement with distribution in singular

Answer: The revision was made.

L 117 – ‘sulfur fungus’   - use scientific name

Answer: The revision was made.

L 119 -  ‘obtained’ ? able to be extracted from the tissue??

Answer: The revision was made.

L 129 -  ‘affect on the growth’  --  delete the ‘on’

Answer: The revision was made.

L 132 -  Table 2 .  (1) mushrooms are not in any order (2) Please spell out the genera (3) Optimized Ca content need some explanation in footnote; contents are in 4 units (4) Calcium enriched needs explanation in footnote – mg/100 g of what?? (5) I would change title ‘ … in some edible mushrooms’

Answer: Good suggestion! The table 2 was carefully corrected according to your suggestion.

L 135 – 136  ‘provides’ to ‘provide’ --- agreement with singular subject

Answer: The revision was made.

L 143 – ‘same group’ -  unclear

Answer: The clarification was made.

L 145 -  “might’  -  words shows doubt.  Is this speculation?

Answer: No, the revision was made.

L 146 –‘demonstrate’ to ‘demonstrated’

Answer: The revision was made.

L 156 ‘could’ – is this speculation or fact? ; same with L 159, L 171

Answer: The revision was made.

L 160 – ‘pattern’  -  unsure of the references (not bibliography) in the text

Answer: The revision was made.

L 162 -3 – ‘in general … mushrooms’ should it not read ‘Edible mushrooms are more responsive to organic Ca salts’ ???

Answer: The revision was made according to your suggestion.

L 165 ‘exhibited the strongest ability’ - rephrase

Answer: The revision was made.

L 185, 188 (2022) or (2017)  – authors have switched referencing style; Sulfur fungus – Latin name please

Answer: The revision was made.

L 188 – ‘study of Yang’ – change ‘of’ to ‘by’

Answer: The revision was made.

L 194 – ‘are one of rapidly …’ to ‘are rapidly growing …’

Answer: The revision was made.

L 196 -  change ‘impelled us’ ‘impells us’  or ‘drives us’ or ‘stimulates us’

Answer: The revision was made.

L 198 – “As showed in Table 3,”  wordy; place word (Table 3) in the sentence

Answer: The revision was made.

L 199 – 200 -  I don’t think the authors need this sentence.

Answer: The sentence was deleted.

L 201 -   Why this table?  Why not incorporate it into a separate paragraph that begins with “The addition of Ca ions …”

Answer: In order to explain effect of calcium enrichment on nutritional value of edible mushrooms to the readers more clearly, the table 3 was preserved in corrected manuscript.

L 205 – “in different” to “differently”

Answer: The revision was made.

L 207 – delete “not”  - “not” is not required if “no” is earlier

Answer: The revision was made.

L 207 – ‘the maximum of mycelium’ to ‘maximum mycelial”

Answer: The revision was made.

L 208 – ‘are’ to ‘were’

Answer: The revision was made.

L 208 – ‘In the work studied by Ji et al. (2017),’  - inconsistent reference style

Answer: The revision was made.

L 215 – delete ‘obviously’

Answer: The revision was made.

L 221 – “scavenge free radicals activity” - I think the authors want to say “free radical scavenger activity”

Answer: The revision was made according to your suggestion.

L 222, 223 “Normally”, “Not surprising”  - wordy, delete

Answer: they were deleted.

L 224 – “normally” - this usage prepares the reader for a contrast like “however”.  I don’t see it. So delete the word.

Answer: The word was deleted.

L 226 – starting sentence with element abbreviation is not recommended.  Use full word – Selenium

Answer: The revision was made.

L 230 – “could” – is this conjecture? Probably authors are saying “mixture modified”

Answer: Your suggestion was adopted.

L 233 – ‘phenolics’ to ‘phenolic’

Answer: The revision was made.

L 235 – unclear relationship or reference to something earlier

Answer: The revision was made.

L 240   ‘was in favourable’   - unclear

Answer: The clarification was made.

L 251 ‘could reduce” to ‘reduced’

Answer: The revision was made.

L 252 ‘initially’  - this word presupposes a contrast.  I see nothing.  Delete it.

Answer: The revision was made.

L 253 – ‘suggested’ to ‘suggest’

Answer: The revision was made.

L 257 – ‘literature .. studied’ to ‘have documented’

Answer: The revision was made.

L 258 – ‘with’  to ‘to’

Answer: The revision was made.

L 259 – ‘therefore, the illustration of’ to ‘Investigating the’

Answer: The revision was made.

L 259 – ‘about’ to ‘of’

Answer: The revision was made.

Section 4 “The mechanisms ….  needs to be re-organized (re-written).  Paragraph 1 mostly is research desired. There are English flow issues and grammar in it. It cannot be cut and pasted directly elsewhere.  Paragraph 2 is a review as stated for the paper.

Answer: Good suggestion! The revision was made. Please see the paper.

L 264 – ‘mushrooms, afterwards, ‘ to “mushrooms. Afterwards,’  -  run-on sentence

Answer: The revision was made.

L 267 – reference style issue

Answer: The revision was made.

L 269 – 271 – rewrite into two sentences and clarify

Answer: The clarification was made.

L 272 – ‘was improved’ to ‘was increased’

Answer: The revision was made.

L 273 -4 – unclear

Answer: The clarification was made.

L 275 -  reference style problem

Answer: The revision was made.

L 277 -  “the authors”  to “these authors” for clarity

Answer: The clarification was made.

L 279 – reference style problem

Answer: The revision was made.

L 279 -  what is the antecedent of “it” – needs clarification

Answer: The clarification was made.

L 280 –  this is a run-on sentence.  Divide into two.

Answer: The revision was made.

L 281 – recently problem.  Also reference style problem.

Answer: The revision was made.

L 281 – ‘absorbed’ to ‘absorb’

Answer: The revision was made.

L 282 -  ‘under’ to ‘with’

Answer: The revision was made.

L 283 – don’t start sentence with abbreviation, no comma after mushrooms.

Answer: The revision was made.

L 290 – don’t start sentence with abbreviation

Answer: The revision was made.

L 298  - ‘regarding on’ to ‘regarding’

Answer: The revision was made.

L 300 – 302 – unclear; rewrite

Answer: The sentence was re-written.

L 305  ‘choose’ to ‘choice’

Answer: The revision was made.